# Corticotropin-Releasing Hormone: Biology and Therapeutic Opportunities

**DOI:** 10.3390/biology11121785

**Published:** 2022-12-08

**Authors:** Alessandra Caruso, Alessandra Gaetano, Sergio Scaccianoce

**Affiliations:** Department of Physiology and Pharmacology “V. Erspamer”, University Sapienza of Rome, 00185 Rome, Italy

**Keywords:** corticotropin-releasing hormone, CRH, stress, reproduction

## Abstract

**Simple Summary:**

Early neuroendocrine studies on corticotropin-releasing hormone (CRH), or corticotropin-releasing factor (CRF), were focused on investigating its role in regulating the hypothalamic–pituitary–adrenal axis. In the following years, the characterization of CRH receptors and the availability of specific CRH agonists and antagonists have provided evidence that CRH plays a role in the regulation of several biological systems, as well as in reproduction, neuropsychiatric, gastrointestinal, and immune disorders and in the development of tumors. Further elucidation of the physiology of CRH will facilitate characterization of its role in human pathophysiology and exploit the potential of ligands for CRH receptors as novel therapeutic targets.

**Abstract:**

In 1981, Wylie Vale, Joachim Spiess, Catherine Rivier, and Jean Rivier reported on the characterization of a 41-amino-acid peptide from ovine hypothalamic extracts with high potency and intrinsic activity stimulating the secretion of adrenocorticotropic hormone and β-endorphin by cultured anterior pituitary cells. With its sequence known, this neuropeptide was determined to be a hormone and consequently named corticotropin-releasing hormone (CRH), although the term corticotropin-releasing factor (CRF) is still used and preferred in some circumstances. Several decades have passed since this seminal contribution that opened a new research era, expanding the understanding of the coding of stress-related processes. The characterization of CRH receptors, the availability of CRH agonists and antagonists, and advanced immunocytochemical staining techniques have provided evidence that CRH plays a role in the regulation of several biological systems. The purpose of this review is to summarize the present knowledge of this 41-amino-acid peptide.

## 1. The Physiology of CRH

A 41-amino-acid peptide with high potency and intrinsic activity stimulating the secretion of adrenocorticotropic hormone and β-endorphin by cultured anterior pituitary cells was characterized from ovine hypothalami by Vale’s group in 1981 [1]. In the following years, the sequence of this peptide, which was previously referred to as Vale’s corticotropin-releasing factor (CRF) or ovine-CRF, was determined, and consequently, the nomenclature referring to the peptide was changed from CRF to corticotropin-releasing hormone (CRH).

CRH is derived from a 192-amino-acid preprohormone and is present in the greatest abundance in parvocellular neurons in the paraventricular nuclei, the axons of which project to median eminence. Immunoreactive CRH is also detectable in human maternal plasma, where it increases progressively with gestational age, reaching peak concentrations during the third trimester of pregnancy [2]. The placenta is a major producer of peripheral CRH and, after the hypothalamus, is the major source of circulating CRH. A binding protein (BP) specific to CRH is present in circulation; CRH-BP is a highly conserved protein that binds also urocortin 1, urotensin 1, and sauvagine and has been hypothesized to exert an inhibitory effect, preventing the activation of CRH receptors [3]. Mechanisms governing CRH release have been thoroughly investigated, particularly with respect to its role in the control of the hypothalamic–pituitary–adrenal (HPA) axis [4]. GABAergic interneurons located in the periventricular region of the hypothalamus exert inhibitory control over CRH release under basal conditions [5,6]. As a result of stress, signals depart from the higher nerve centers that reach the neuroendocrine cells of the paraventricular nucleus (PVN) of the hypothalamus, stimulating the production and the release of CRH in pituitary portal circulation. The rapid CRH activation required in response to stress is accomplished by noradrenergic and adrenergic monosynaptic pathways with direct projections to the PVN [7]. The CRH released into portal circulation reaches the adenohypophysis and, by acting on corticotropic cells by binding with specific receptors, promotes the release of adrenocorticotropic hormone (ACTH). Once released into systemic circulation, ACTH reaches the cortical area of the adrenal glands, where it regulates the synthesis and secretion of adrenal glucocorticoid hormones. CRH receptors have been the subject of extensive reviews [8,9,10,11]. CRH acts by activating two receptors of class B1 GPCRs, CRHR1, and CRHR2. The CRHR1 gene has been shown to express nine subtypes (α, β, c, d, e, f, g, h, and v_1), which are produced by the alternative splicing of exons 3–6 and 10–13. Three known subtypes (α, β, and γ) are expressed by the CRHR2 gene and produced by alternate 5′ exons [12,13].

CRHR1 is the major subtype in anterior pituitary corticotrope cells, where it mediates the stimulatory actions on ACTH secretion. CRHR1 is also present in brain cortical areas, in the cerebellum, and in the limbic system. Several splice variants of CRHR2 have been reported, which are present in subcortical brain regions (i.e., dorsal raphe nuclei of the midbrain, ventromedial hypothalamus, and nucleus of the solitary tract) and in the periphery (i.e., in the heart and skeletal muscle) [14]. Interestingly, both receptors have been identified in the myometrial smooth muscle, where they play a role in myometrial contractility and quiescence during pregnancy and labor [15,16]. Moreover, the presence of both CRHR1 and CRHR2 has been documented in human epidermal and dermal cells. In particular, CRHR1 has been detected in the cellular populations of the epidermis, dermis, and subcutis, whereas CRHR2 is expressed in hair follicles, sebaceous and eccrine glands, muscle, and blood vessels [17,18]. Finally, a third receptor with distinct ligand-binding characteristics and tissue distribution has been reported in catfish pituitary and urophysis systems [19].

## 2. CRH Actions

Several excellent reviews on the main role of CRH as the central regulator of the HPA axis are available in the literature [20,21,22]. The physiological mechanisms related to the CRH control of ACTH secretion are described above. However, arginine-vasopressin (AVP) has been proposed to play a coagonistic role in the control of ACTH secretion [23,24]. CRH has been demonstrated to play a role in modulating other signaling pathways, conferring the ability of this peptide to exert indirect actions [25]. Whole-cell patch-clamp recordings in primary cultured hippocampal neurons indicated that CRH inhibited N-methyl-d-aspartate (NMDA)-induced currents in a dose-dependent manner through CRHR1 [26]. CRHR1 also mediates activation in both rodent and human pancreatic islet cells, stimulating β-cell proliferation and insulin synthesis in a glucose-dependent manner [27,28]. As suggested by Brar [25], it can be speculated that the insulinotropic actions of pancreatic CRHR1 opposes the activation of CRHR1 on anterior pituitary corticotropes, which, through the ACTH-induced release of glucocorticoids, functionally antagonize the actions of insulin. Moreover, in an animal model of persistent inflammatory pain, a recent report indicated that central CRHR1 activation elicited potent antinociceptive effects. Finally, the colocalization of opioid peptides with CRHR1 in neurons within pain-relevant brain areas strengthens the hypothesis of the involvement of CRH in the control of the opioid system [29]. Autoradiographic studies in rat brains have demonstrated that the lateral septum, posterior colliculus, and sensory trigeminal nucleus display solely CRHR2, which may contribute to the co-ordination of the integrated stress response [22]. Additional actions of CRHR2 are described below.

## 3. CRH in the Reproductive System

CRH and its receptors have been identified in the female reproductive system [30]. Ovarian CRH is principally localized in theca cells surrounding the follicles, in the luteinized cells of the stroma, and in the cytoplasm of the ovum, where it inhibits ovarian steroidogenesis [31,32]. Uterine epithelial cells are the main source of endometrial CRH [33]. Endometrial CRH is expressed throughout the menstrual cycle, reaching a peak during the luteal phase. This menstrual-phase-dependent CRH increase serves as the basis for the hypothesis that CRH might be involved in processes such as decidualization and embryo implantation [34]. Regarding decidualization, it has been proposed that progesterone induces the transcription of the CRH gene in the endometrial stroma and that this CRH–progesterone interplay actively participates in the decidualization of endometrial stroma [33]. It is well known that during pregnancy, the placenta and fetal membranes produce substantial quantities of CRH, especially during the second and third trimesters of pregnancy. Interestingly, the increase in maternal plasma CRH concentrations with advancing pregnancy correlates with a concomitant reduction in the concentrations of CRH-BP, which results in an increase in free, bioavailable CRH at the onset of parturition [35]. A clinical study recently suggested that placentally derived CRH positively modulates fetal liver blood flow [36]. Moreover, placental CRH induces dilation of uterine and fetal placental vessels through nitric oxide synthetase activation [37,38]. The effects of CRH and CRH-related peptides (i.e., urocortin II and urocortin III) have been studied with regard to the activation of the nitric oxide/cGMP pathway in normal and pre-eclampsia placentas [39]. In pre-eclampsia placental explants, the CRH-induced cGMP response was found to be reduced concomitantly to a downregulation of a specific CRHR2 isoform, namely R2β. Overall, these data suggest that dysregulation of CRH signaling may contribute to the altered vascular resistance balance of this clinical condition. In pre-eclampsia, trophoblast invasion of the interstitial uterine compartment is frequently shallow. Anomalies in the process of trophoblast invasion may induce abnormal placentation [40]. Bamberger and colleagues [41] demonstrated that through CRHR1 activation, CRH inhibits trophoblast invasion, an effect that may be relevant in the pathophysiology of pre-eclampsia and placenta accreta. Decreased expression of CRHR1 has been observed in both pre-eclampsia and intrauterine growth retardation, reinforcing the hypothesis that CRH plays a role in controlling vascular resistance [42].

During pregnancy, adrenal function undergoes significant changes, and the management of adrenal insufficiency in pregnancy requires rapid diagnosis [43]. As mentioned above, plasma CRH concentrations increase during pregnancy [35]. An early increase in peptide plasma concentrations was recently reported in women who had experienced prior preterm birth, suggesting that measurement of CRH during the second trimester may identify women at risk of preterm delivery [44]. Moreover, it has been proposed that in term pregnancies, placental CRH induces myometrial contractility through prostaglandin production from the decidua and fetal membranes [45], thereby potentiating the contractile effects of prostaglandins and oxytocin on the myometrium [46]. An additional action of the increase in placental CRH during the last phase of pregnancy has been attributed to a stimulatory tone on the production of dehydroepiandrosterone sulfate (DHEA-S) by the fetus. A physiological role of this increase in DHEA-S could be related to its conversion to estrogen in the placenta, thereby participating in the hormonal control in this phase of pregnancy [47]. Finally, CRH has been found in the testis of some species (i.e., rat and ovine) [48], but its role has not been completely elucidated to date.

## 4. CRH in Neuropsychiatric Disorders

Although the precise etiopathological mechanism involved in or triggering affective disorders has not been completely clarified, several lines of evidence strongly suggest that stress and stress-related homeostasis derangement are involved in the causation and development of depression and other psychiatric disorders. Moreover, hypersecretion of cortisol at baseline and the reduced feedback action of dexamethasone (or in the combined CRH–dexamethasone test) are frequently observed among depressed patients [49,50]. This hypothesis has constituted the rational basis to investigate the role of CRH, as well as CRH antagonists, in these pathologies, although none of the drugs acting as CRH antagonists have been clinically applied to date [51,52]. α-helical CRH 9–41 (a competitive CRHR2 antagonist) was synthesized in 1984. The first report on its ability to act as a competitive CRH antagonist both in vitro and in vivo was published in the same year [53]. The first clinical trial investigating the effects of α-helical CRH 9–41 in healthy subjects was performed in 1996 [54]. Results indicate that the synthetic antagonist reduced ACTH and cortisol levels without affecting blood pressure, glucose, or electrolyte imbalances. Although there is currently no consensus on a suitable biological biomarker(s) underpinnings depression [55], it has been reported that the cerebrospinal fluid of depressed patients contains elevated levels of CRH [56]. However, conflicting results with respect to cerebrospinal fluid CRH levels in depressed patients have been reported. These differences may be associated with diurnal variations in CRH release, the subtype of depression, and/or whether the patients were previously under antidepressant therapy [57]. Nevertheless, increased cerebrospinal fluid CRH concentrations have been reported in patients with post-traumatic stress disorders [58] and in subjects affected by anorexia nervosa [59]. The hypothesized CRH central hyperactivity could be responsible for the reduced receptor density, which has been detected in the frontal cortex of depressed patients [60]. However, it is difficult draw a conclusion as to whether CRH per se is involved in the pathologies of affective disorders. Nevertheless, the frequently reported hyperactivity of the HPA axis observed in these psychiatric pathologies has contributed inspired the study of the potential therapeutic effect of pharmacological antagonism to CRH receptors. In addition, a direct, HPA-axis-independent involvement of CRH in mood disorders has long been hypothesized based on the finding that central CRH signaling is responsible for anxiety-like behavior [61]. Recently, several studies have been conducted to ascertain the effects of CRH antagonists in animal models of depression. Rodents (rats and mice) are the most investigated species, and the methodologies used to induce an endophenotype of depression vary substantially between laboratories. Nevertheless, the key features induced by experimental manipulations frequently converge in anxiety, sleeplessness, decreased sucrose intake in the sucrose preference test, decreased sexual interest, and psychomotor agitation, among other manifestations. Several compounds with specific characteristics to selectively antagonize type 1 receptors and type 2 receptors, as well as non-selective receptor recruitment, have been synthesized [62]. Among them, antalarmin (CP-156,181), a non-peptide CRHR1 antagonist with an affinity in the low nanomolar range, and the closely related drug, SSR125543A (crinecerfont), were reported to reduce the duration of immobility in rates in forced swim tests, indicating an antidepressant-like effect of these compounds [63]. The forced swimming test has been used to investigate the potential antidepressant-like effects of four CRHR1 antagonists and the corresponding effect on swim-induced HPA activation [64]. Results indicate that antalarmin and R121919 did not modify the immobility in the forced swim test, although all three compounds decreased the surge of plasma ACTH induced by the stressful condition linked to the experimental manipulation. It is tempting to hypothesize that these compounds, acting on pituitary corticotropes to efficiently suppress ACTH release, have a low CNS penetration, therefore failing to induce antidepressant activity in the forced swim test. On the contrary, compound SC-241/LWH-234 reduced immobility in the forced swim test without altering the swim-stress-induced ACTH increase, although it blunted the corticotropin response to restraint stress. The latter findings, which suggest a dichotomy between the CRH control of HPA axis activation and its postulated role in reducing a behavioral profile indicative of depression, may provide additional relevant clues on this subject. However, to the best of our knowledge, no other studies have been published on this promising compound. Pexacerfont is an orally active CRHR1 antagonist that has been clinically evaluated for the treatment of stress- and dependence-induced alcohol seeking and anxiety disorders. Although on the basis of the drug levels in the cerebrospinal fluid, it was a 90% central CRHR1 occupancy was predicted, pexacerfont treatment had no effect on alcohol craving, emotional responses, or anxiety [65]. Another orally active compound, verucerfont, has been evaluated for its ability to suppress HPA-axis activation and reduce stress-induced alcohol craving in alcohol-dependent patients. Again the reported findings obtained do not support the clinical efficacy of CRHR1 blockade in stress-induced alcohol craving and relapse [66]. CRH-dependent HPA-axis overactivation has been investigated in relation to neurological disorders such as Alzheimer’s disease [67,68]. Central antagonism of CRH receptors remains an interesting research field in neuropsychiatric disorders. Whereas the pharmaceutical industry should develop blood–brain-barrier-penetrating, selective CRHR antagonists, preclinical research is required to identify suitable animal models of depression because, for example, the forced swimming test, although it has demonstrated some validity in predicting the antidepressant potential of compounds, has been repeatedly questioned as a test to phenotype animals as ‘depressed’ [69]. With respect to the need for a new selective CRHR antagonist, a thiazolo [4,5-d] pyrimidine CRHR1 antagonist was recently synthesized [70], which is characterized by increased binding affinity relative to a standard CRHR (i.e., antalarmin) and may disclose advance some interesting pharmacotherapeutic opportunities.

## 5. CRH in Gastrointestinal Disorders

CRHR1 and CRHR2 have been detected throughout the gastrointestinal tract in various cell types, including endocrine (enterochromaffin cells), neuronal (the enteric nervous system), and immune cells (eosinophils, mast cells, and T-helper lymphocytes in the lamina propria), indicating a local role of the CRH systems within the gut [71]. Chronic stress can affect intestinal barrier dysfunction and impair host defense mechanisms mediated by CRH and mast cells. Evidence of the pivotal role of CRH includes the finding that chronic peripheral administration of this peptide causes colonic barrier dysfunction similar to prolonged psychological stress exposure [72]. Moreover, social stress in rats causes urinary retention and abnormal urodynamics, an effect that is prevented by the antagonist NBI-30775 [73]. These findings have constituted the basis for several preclinical studies with the goal of ascertaining the potential pharmacotherapeutic role of CRH antagonists as a treatment for abdominal and pelvic disease [74]. Inflammatory bowel disease (IBD) and irritable bowel syndrome (IBS) are the most-tested disease models in preclinical studies to ascertain the effectiveness of CRH antagonism. Stressful conditions experienced in infancy might constitute a risk factor for gastrointestinal disease in adulthood [75]. Early maternal separation is an experimental model frequently used to study IBS and IBD. In mice, maternal separation has been demonstrated to induce the activation of CRHR1 which promotes intestinal inflammation, modulates gut permeability, and affects the composition of the intestinal microbiota [76]. Interestingly, CRHR2 signaling activates intestinal stem cells, contributing to intestinal injury repair [76]. Collectively, these results suggest that inhibiting the action of CRHR1 and promoting the activity of CRHR2 could represent a pharmacotherapeutic strategy for gut injury in preterm infants. Moreover, a recent report indicated that in an IBD mouse model, CRH promotes inflammatory bowel disease by enhancing intestinal macrophage autophagy [77].

## 6. CRH and Inflammation/Immune Response

Given its role as a principal regulator of the activity and reactivity of HPA axis, it is reasonable that CRH administration, as stress application, suppresses the immune system indirectly via glucocorticoid and/or sympathetic system-mediated mechanisms. Moreover, during inflammation, cytokines TNF alpha, IL-1, and IL-6 stimulate hypothalamic CRH and/or AVP secretion as a means of preventing or blunting the inflammatory reaction. Furthermore, CRH receptors have been detected in peripheral sites of the immune system; CRH was found to promote several immune functions in vitro; and mast cells, monocytes/macrophages, neutrophils, and other types of immune cells express both types of CRH receptor [78,79]. It has been hypothesized that CRH produced in peripheral inflammatory sites, contrary to its systemic indirect immunosuppressive effects, acts as an autocrine or paracrine inflammatory cytokine [80,81]. A question remains: How does CRH directly influence immune response? Chandras and colleagues conducted a study in human monocytic THP-1 cells, a commonly used model to estimate modulation of monocyte and macrophage activities, and demonstrated that CRH stimulates the phosphatidylinositol 3-kinase (PI3K)/Akt and ERK1/2 pathways through CRHR2 activation. In particular, PI3K/Akt activation promotes cell survival through the activation of the antiapoptotic factor Bcl-2, whereas ERK1/2 stimulation results in the upregulation of IL-8 expression, an effect inhibited by the CRH-induced activation of PI3K/Akt [82]. The expression of CRHR2 was recently reported on splenic murine B cells [83]. In the same study CRH was demonstrated to decrease the viability of splenic B cells, rendering them more sensitive to death induced by CRH produced in response to stress. Overall, these findings expand our knowledge of the CRH-dependent mechanisms of the immune system and the humoral immune response and can serve as a valuable reference for the development of pharmacotherapeutic agents for inflammatory diseases.

## 7. CRH in Cancer Biology

Several lines of evidence have suggested a role of CRH in cancer. CRH and CRH-related peptides (e.g., urocotins), as well as specific receptors, have been detected in several tumor tissues, serving as a rational basis not only to hypothesize that CRH may be involved in cellular proliferation, migration, invasion, and apoptosis but also offering potential pharmacological targets. Peripheral CRH has been detected in ovarian carcinoma [84]. Using both ovarian cancer tissue and two ovarian cancer cell lines, Minas and colleagues [85] demonstrated that CRH contributes to the immune privileges of ovarian tumors. Regarding the molecular mechanism, the authors provided evidence that CRH acts on ovarian cancer through CRHR1, which upregulates the expression of Fas ligands (FasL), potentiating the ability to induce Fas-mediated apoptosis. Therefore, it is tempting to hypothesize that CRHR1 antagonists could constitute a rational pharmacological target to treat ovarian cancer. An additional correlation between CRH and FasL activation has been reported in human cervical carcinoma cells (HeLa). When HeLa cells were exposed to CRH, a significant increase in both FasL transcription and FasL translation was detected, reinforcing the hypothesis that CRH is involved in tumor immunoescape [86]. The involvement of CRHR1 in the development of colitis-associated cancer (CAC) was investigated in *Crhr1*-deficient mice [87]. The results indicate a proinflammatory and protumorigenic role of CRHR1 in inducing macrophages to release proinflammatory cytokines and activating NFkB–BCL2 and STAT3 signals to reduce apoptosis and enhance proliferation. Moreover, the potential role of CRH/CRHR1 signaling has been investigated in two kinds of colon cancer cell lines (i.e., DLD1 and HCT116 cells). The results indicate that, through CRHR1 activation, CRH promotes cell growth through NF-κB/IL-6/JAK2/STAT3 signaling and tumor angiogenesis through NF-κB/VEGF signaling [88]. In the same study, it was also demonstrated that CRHR1, CRH, and urocortin were highly expressed in carcinomatous tissues compared with normal and pericarcinomatous tissues.

Ectopic Cushing’s syndrome caused by an ACTH-CRH-producing pheochromocytoma is an extremely rare clinical condition. Using single-cell RNA sequencing, Zhang and colleagues identified a novel multifunctionally chromaffin-like cell type with high expression of both pro-opiomelanocortin (the precursor of ACTH) and CRH [89]. They also detected specific expression of galanin (GAL) and proposed GAL as a candidate marker to detect ACTH-CRH-producing pheochromocytoma in suspected cased of Cushing’s syndrome.

## 8. Conclusions

Although several decades have passed since the characterization of CRH, ongoing research is continuously suggesting new and unexpected roles for this peptide. Moreover, preclinical data have convincingly supported the therapeutic potential CRH receptor antagonists/agonists, although the specific patient profile that may benefit from such pharmacological treatment remains to be identified, as is the case for the role of CRH in the neurobiology of depression. Given that more than 30% of people affected by this pathology suffer from treatment-resistant depression [90], the opportunity to investigate the effects of CRHR1 antagonists in this specific cohort of patients is of particular interest. Finally, new pharmacodynamic approaches that confer to drugs the ability to contemporaneously behave as an agonist and as an antagonist with respect to receptor occupation have potential for application in conditions in which inhibition of CRHR1 activation and CRHR2 stimulation could represent a pharmacotherapeutic strategy for example, in gut injury in preterm infants.

## Data Availability

Not applicable.

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
