# Peer review of "Corticotropin-Releasing Hormone: Biology and Therapeutic Opportunities"

_biology, 2022, doi:10.3390/biology11121785_

Round 1

Reviewer 1 Report (Previous Reviewer 2)

The author addresses all my questions appropriately.

Author Response

No actions required

Reviewer 2 Report (New Reviewer)

Lines 68-69, 180

CRH1R and CRH2R are wrong. It is correct CRHR1 and CRHR2

Lines 73 -92

In the paragraph on biological actions of CRH, the authors excellently expanded the information requested, corresponding to the participation of the CRH receptor. However, they only targeted the CRHR1 receptor, and I think they should include information on CRHR2 as well.

In the rest of the document, the authors included the appropiate information that was required by the reviewers.

Additional comments:

The authors did an excellent review job in addressing the direct and indirect roles of CRH and its receptors in various systems, as well as, its implications in therapeutic as they describe, so what makes information relevant. In addition, they adequately attending each of the observations that the reviewers marked. However, it would be important for them to include information on CRHR2 in the paragraph on the biological actions of CRH, just as they did for CRHR1, this will make the information of the participation of the receptors of CRH more clear, and finally to review the nomenclature of the receptors because there are some errors. 

Author Response

Response to Reviewer #2

Lines 68-69, 180

Typos have been corrected.

Lines 73-92

Information on CRHR2 have been included.

Reviewer 3 Report (New Reviewer)

1) It is unfortunate that the US Endocrine Society has insisted on naming CRF41 as CRH.  This is misguided on three counts. First CRF41 is not the only releasing hormone for ACTH

Second,  its actions extend way beyond the CRH function. Third in recognizing this, IUPHAR recommend CRF1 etc as  the name of the receptors for CRF41 like peptides.  At the time of discovery Vale and coworkers suggested using the name alarmin. This should be discussed in the text. Better still don't use CRH use CRF41.

2) English usage should be improved throughout the paper.

3) The section headings are somewhat confusing.  Is the only “physiology” of CRF41 within the hypothalamus? Are not all actions of CRF41 biological (line 73)?

4) It is unclear what the sections highlighted in yellow are.

5) line 88 – NMDA is not produced in the body.

6) Line 183 The forced swimming test is not regarded as a good predictive model for antidepressant efficacy.

7) Ultimately, there are no useful CRF receptor modulating drugs in clinical trials.  Moreover, the recurrent idea that an overactive HPA axis somehow facilitates depression has been repeatedly tested but remains unproven.

Author Response

1) We gratefully acknowledge reflection on the proper nomenclature proposed by the Reviewer. However, i): in the manuscript we stated that “arginine-vasopressin (AVP) has been proposed to exert a co-agonistic role on the control of ACTH secretion [Rotondo et al., 2016; Antoni, 1993]; ii) we hope that the peptide’s functions not related to ACTH secretion are reported in the manuscript; iii) we were aware that in 2003 IUPHAR recommended the use of CRF nomenclature, but later, maybe under the influence of the Endocrine Society, CRH is the term mostly used. Finally, we do remember the early days after Vale’s group characterization. In 1982 one of us was working at the Sherrington School of Physiology with the late Professor Mortyn T. Jones, and the (very small amount of the) peptide was named “Vale’s CRF”.  Nevertheless, in the abstract of revised manuscript, as well as in the Simple Summary, the use of the acronym CRF is reported.

2) The manuscript was checked by a native English speaker. Some corrections have been made in the revised manuscript.

3) The word “biological” (line 73) has been deleted.

4) Removed all the yellow highlighted sections.

5) Line 88 has been corrected.

6) We do agree with the Reviewer that the forced swimming test is not regarded as a good predictive model for antidepressant efficacy. For this reason, we stated that: “the forced swimming test, although it has demonstrated some validity in predicting a compound's antidepressant potential, has been repeatedly questioned as a test to phenotype animals as 'depressed'”. In line with this consideration, we have referred to a recent study by Molendijk and de Kloet who analytically analyzed this experimental model (i.e., Molendijk, M. L.; de Kloet, E. R. Forced swim stressor: Trends in usage and mechanistic consideration. Eur. J. Neurosci. 2022).

7) In the present manuscript, we have never stated that HPA axis hyperactivity is the main causative factor responsible for depression. We have reported studies indicating a correlation, not a causation. In addition, we stated: “This hypothesis has constituted the rational basis to investigate the role of CRH, as well as CRH antagonists, in these pathologies (i.e., affective disorders) although at present none of the drugs acting as CRH antagonists reached the clinic”.

This manuscript is a resubmission of an earlier submission. The following is a list of the peer review reports and author responses from that submission.

Round 1

Reviewer 1 Report

no comments

Reviewer 2 Report

The author summarized the function and action mechanism of CRH and its associated pre-clinical studies aiming to develop the HPA axis as a therapeutic target for multiple diseases. However, as a review paper, the author should better organize the information and make them flow better. The current version is hard for the reader to follow and extract the key information. 

1. In line 31, the author forgets to reference the paper to support their claim. 

2. The author excellently summarizes the physiological function of CRH. However, in the second part-CRH biological actions, it lost focus. We believe the author would like to introduce the expression profile of the CRH receptor across the body and their associated physiological and pathological function. First, we recommend the author should introduce that the CRH function could exert both directly and indirectly, as the author did in a later paragraph. The author only introduces the CRH expression in the reproductive system under the title of "CRH biological actions," which is incomplete. As introduced by the author, the CRH biological function is involved in many systems, including the reproductive, neurological, enteric, and immune systems. 

3. In the reproduction system, the CRH directly exerts important functions through the local CRH receptor. However, unlike the neurological system, no related pathological condition is based on the aberrant CRH system. We wonder if there are reports on abnormal CRH signaling that may contribute to pathologies in the women's or men's reproduction system. To our best knowledge, the CRH signaling pathway is related to preeclampsia, abnormal placenta invasion, endometrial growth retardation, and preterm delivery. Please review these functions and related pre-clinical progress.

4. In part 3, the CRH in neuropsychiatric disorders", the author like to put conflict and limitations of a theory before a full introduction (line 124, line 150-155), which is very destructive. We suggest the author complete the study's introduction before pointing out their limitation. Like in CRH1R antagonists, the author could introduce their target, pharmacological mechanism,  and result. Since the results show that the administration did not modify the animal behavior, the author could introduce the study's limitation but not vice versa. Similarly, the author introduces the role of the HPA axis in abnormal behavior profiles, followed by the pre-clinical model's failure. However, the author didn't give their own opinion or limitation analysis. 

5. The author should abbreviate or use other shorter version names of those CRFR1 antagonists. 

6. Are CRH or CRH receptors expressed in the gastrointestinal system, or do they also exert their function indirectly like in the immune system?

Reviewer 3 Report

The review article from Caruso et al. focused on the functional role of corticotropin-release hormone. Overall, this article provided a good review of the basic information about CRH. However, most of the content are superficial. More importantly, even the title of this article included the term “therapeutic opportunities”, but the review and discussion about the use of CRH (agonist and/or antagonist) is minimal, and most of the information is about neuropsychiatric disorders. Also, the role of CRH in cancer biology is missing.

Major concerns:

1.     Some sentences are poorly written, for example, Line 27: “Like other genes, the CRH gene is very conservative in vertebrate species.” What are the “other genes”? Also, the sequence conservation of the genes should be described precisely, such as using the % homology or identity of DNA or amino acid sequence.

2.      Like mentioned above, at Line 38, please define what is “an ancient” protein.

3.      Line 29-13, the production of protein is too very known and not necessary to describe in the article.

4.     Line 55-59, a detailed table/graphic summary to compare the structure/sequence of CRHR1/2 and their isoform/subtype is recommended.

5.     In addition, is there any functional different of the CRHR1 vs CHRH2 and their isoforms?

6.     Section 2 “CRH biological actions” is only focused on reproductive biology, I suggest changing the title of this section.

7.     As mentioned, the role of CRH and its receptor in cancer biology should be added. Especially, the recent finding about the single-cell transcriptome of ACTH and CRH secreting pheochromocytoma should be discussed.

8.     Line 223-224, authors mentioned, “Unfortunately, in some instances phase I trials have been disappointing”. Authors should provide more information. Indeed, to make this article fit the title, authors should have a section to review the agonist and antagonists of CRHR1/2 as well as the progress of related clinical trials.

9.     Overall, this review article only cited around three references on/after 2020. More updated information about this area should be included.